# Association between Phenotypes of Antimicrobial Resistance, ESBL Resistance Genes, and Virulence Genes of *Salmonella* Isolated from Chickens in Sichuan, China

**DOI:** 10.3390/ani13172770

**Published:** 2023-08-31

**Authors:** Gang Shu, Jianyu Qiu, Yilei Zheng, Lijen Chang, Haohuan Li, Funeng Xu, Wei Zhang, Lizi Yin, Hualin Fu, Qigui Yan, Ting Gan, Juchun Lin

**Affiliations:** 1Department of Basic Veterinary Medicine, Sichuan Agricultural University, Chengdu 611130, China; dyysg2005@sicau.edu.cn (G.S.); qiujy2018@163.com (J.Q.); lihaohuan7@163.com (H.L.); funengxu@sicau.edu.cn (F.X.); zhangwei26510c@126.com (W.Z.); yinlizi@hotmail.com (L.Y.); fuhl.sicau@163.com (H.F.); yanqigui@126.com (Q.Y.); ganting1997@126.com (T.G.); 2Center for Veterinary Sciences, Zhejiang University, Hangzhou 310058, China; yileizheng@zju.edu.cn; 3Department of Small Animal Clinical Sciences, Virginia-Maryland College of Veterinary Medicine, Blacksburg, VA 24061, USA; ljchang@vt.edu

**Keywords:** *Salmonella*, antimicrobial resistance, ESBL genes, virulence genes, MLST, chicken

## Abstract

**Simple Summary:**

*Salmonella* is an important pathogen, which causes a variety of animal salmonellosis and foodborne diseases in humans. In recent years, due to the abuse of antimicrobials, the rapid emergence of resistant *Salmonella*, especially extended spectrum β-lactamase-positive strains (ESBL-producers), has brought great challenges to animal industries and public health safety. The pathogenicity of *Salmonella* depends on a variety of virulence factors coded by genes in chromosomal pathogenicity islands (SPI) and plasmids. The antibiotic resistance and pathogenic potential of *Salmonella* have been studied extensively. However, there are gaps that still exist in our understanding of the relationship between resistance and virulence. In this study, we collected 117 *Salmonella* isolates from diseased chickens in Sichuan Province of China from 2019 to 2021, and investigated antimicrobial resistance (AMR) patterns, including extended spectrum β-lactamase (ESBL) resistance, the prevalence and con-translation of ESBL and virulence genes, and sequence type (ST) among these isolates. The *Salmonella* isolates showed different frequencies of resistance to antimicrobials, 41.03% of which were ESBL-producers. ESBL genes, such as *bla*_CTX-M-55_ (63.29%), *bla*_OXA-31_ (26.58%), *bla*_CTX-M-65_ (20.25%), and *bla*_TEM-1_ (18.99%), were detected. Moreover, the 117 isolates had 11 virulence genes, with frequencies ranging from 29.06% to 100%. There were associations between resistance to cephalosporins and the ESBL genes. The *bla*_CTX-M-55_ showed the largest effect on the resistance and there was also a significant association between the presence of most virulence genes and ESBL genes. All isolates were divided into 11 types. The co-transfer of ESBL and virulence genes was observed in the plasmid conjugation test. From this study, we can conclude that *Salmonella* isolates from chicken were common carriers of ESBLs and multiple virulence genes, and the horizontal transfer played a key role in the dissemination of antimicrobial resistance and pathogenesis. Importantly, the interaction of resistance and virulence factors reinforces the marked pathogenic potential of *Salmonella*.

**Abstract:**

The aim of this study was to explore the association between antimicrobial resistance, ESBL genes, and virulence genes of *Salmonella* isolates. From 2019 to 2021, a total of 117 *Salmonella* isolates were obtained from symptomatic chickens in Sichuan Province, China. The strains were tested for antimicrobial resistance and the presence of ESBL according to the Clinical and Laboratory Standards Institute (CLSI) instructions. The presence of ESBL genes and genes for virulence was determined using Polymerase Chain Reaction (PCR). In addition, Multilocus Sequence Typing (MLST) was applied to confirm the molecular genotyping. Moreover, the mechanism of ESBL and virulence gene transfer and the relationships between the resistance phenotype, ESBL genes, and virulence genes were explored. The isolates exhibited different frequencies of resistance to antibiotics (resistance rates ranged from 21.37% to 97.44%), whereas 68.38% and 41.03% of isolates were multi-drug resistance (MDR) and ESBL-producers, respectively. In the PCR analysis, *bla*_CTX-M_ was the most prevalent ESBL genotype (73.42%, 58/79), and *bla*_CTX-M-55_ showed the most significant effect on the resistance to cephalosporins as tested by logistic regression analysis. Isolates showed a high carriage rate of *invA*, *avrA*, *sopB*, *sopE*, *ssaQ*, *spvR*, *spvB*, *spvC*, *stn*, and *bcfC* (ranged from 51.28% to 100%). MLST analysis revealed that the 117 isolates were divided into 11 types, mainly ST92, ST11, and ST3717. Of 48 ESBL-producers, 21 transconjugants were successfully obtained by conjugation. Furthermore, ESBL and *spv* virulence genes were obtained simultaneously in 15 transconjugants. These results highlighted that *Salmonella* isolates were common carriers of ESBLs and multiple virulence genes. Horizontal transfer played a key role in disseminating antimicrobial resistance and pathogenesis. Therefore, it is necessary to continuously monitor the use of antimicrobials and the prevalence of AMR and virulence in *Salmonella* from food animals and to improve the antibiotic stewardship for salmonellosis.

## 1. Introduction

Avian salmonellosis is a prevalent and detrimental infectious disease caused by bacteria belonging to the *Salmonella* genus. It encompasses various conditions such as pullorum disease, avian typhoid, and avian paratyphoid. These diseases severely impact the health of poultry flocks. Consequently, avian salmonellosis leads to substantial economic losses in the poultry industry [1]. In addition, some types of *Salmonella*, such as *S. enteritidis* and *S. typhimurium*, are the leading causes of foodborne infections through contaminated eggs and chicken products and constitute a potential risk to public health [2].

In recent decades, antimicrobials have been used to prevent and treat bacterial diseases such as salmonellosis. Antimicrobial resistance is an almost inevitable consequence of using antimicrobials in food-producing animals. Studies have found that the rate of multidrug resistance in *Salmonella* has increased from 20–30% to 70% over the past three decades, and the family of cephalosporins is considered to be an essential antibiotic for salmonellosis caused by multidrug-resistant strains [3]. In recent years, extensive and inappropriate use of cephalosporins in the livestock industry has led to the rapid emergence and spread of extended-spectrum β-lactamase (ESBL)-producing bacteria. The recent reports of the frequencies of ESBL-producing Enterobacteriaceae from food animals indicate the situation is serious. Sahan et al. [4] reported that out of 46 *Salmonella* isolates from broilers and cattle, 93.5% exhibited MDR and 58.70% produced ESBLs. In contrast to *Salmonella*, a high prevalence (90.8%) of ESBL-producing *E. coli* was observed in dairy cattle [5].

The ESBLs are generally defined as β-lactamases that can hydrolyze the extended-spectrum cephalosporins and aztreonam, and are inhibited by β-lactamase inhibitors [6]. The most common ESBLs, such as SHV, TEM, CTX-M, and OXA types in Gram-negative bacteria, have also been reported in *Salmonella*, triggering a severe public health concern [7]. It has been proven that plasmids play an important role in the transmission of ESBL genes. Even without antimicrobials, the spread of ESBL genes can occur via plasmid transfer within a bacterial population [8]. Furthermore, resistance plasmids of *Salmonella* can recombine with serotype specific virulence plasmids to form resistance–virulence hybrid plasmids, which can endow *Salmonella* with the ability to survive antimicrobial exposure and disseminate into a new genetic lineage [9]. The high morbidity and mortality observed in the cases of infection with multidrug-resistant pathogens are related to the interplay between resistance and virulence [10].

The pathogenicity of *Salmonella* is closely related to virulence factors, such as *Salmonella* pathogenicity islands (SPIs), encoding the type III secretion systems, which facilitate the invasion of host cells, the bacterial survival and colonization, and systemic infection. Except for SPIs, virulence genes distributed in virulence plasmids (*spvRABCD*), structural virulence factors genes (*agf*, *bfp*, *bcfC*, *fimA*, etc.), and enterotoxin virulence genes (*stn*) are associated with host infections [11]. A previous report indicated that various virulence genes have been detected in avian *Salmonella* spp. in China. The prevalence of virulence-associated genes in these strains, such as *invA*, *sipB*, *prgH*, *ttrC*, *mgtB*, *siiE*, *pipA*, *spvB* and *spvC* (100%), *sopB* (98.8%), and *sopE* (96.4%), was reported to be high [12]. The prevalence of SPI genes (*mgtC*, *siiD*, and *sopB*), *fimA*, and *stn* was 100%, whereas other virulence genes, such as *ssaQ* (89.4%), *avrA* (86.8%), *spvC* (6.6%), and *spvR* (5.3%), were detected in the *Salmonella* isolated from duck meat [13].

In China, there have been various studies of the prevalence of antimicrobial resistance and virulence genes of *Salmonella*. However, there are few studies on whether the development of resistance will affect the pathogenicity of the strain. Whether there is co-transfer of resistance genes and virulence genes in *Salmonella* remains to be studied. Therefore, the purpose of this study was to determine the relationship between ESBL genes and the resistance profile and virulence genes in *Salmonella* isolates from chickens in the Sichuan Province of China, and to assess the co-transfer status of drug resistance genes and virulence genes.

## 2. Materials and Methods

### 2.1. Bacterial Isolates

During the period from 2019 to 2021, there were 117 *Salmonella* isolates obtained from dead or sick chickens on poultry farms located in the Sichuan Province of China. *Salmonella* was isolated following the procedures in the US Food and Drug Administration (FDA) Bacteriological Analytical Manual [14]. In addition, confirmation of 16S rDNA and *invA* genes of isolates was achieved by PCR [15].

### 2.2. Antimicrobial Susceptibility Testing and ESBL Confirmatory Test

The broth microdilution methods were used to determine the antimicrobial susceptibility of 117 *Salmonella* isolates according to the CLSI guidelines [16]. The following antimicrobials were tested in this study: ampicillin (AMP), cefazolin (CFZ), ceftriaxone (CRO), cefotaxime (CTX), gentamicin (GEN), kanamycin (KAN), doxycycline (DOX), nalidixic acid (NAL), ciprofloxacin (CIP), and colistin sulfate (COL). The concentration range of ciprofloxacin was from 0.015 to 16 μg/mL, and the concentration range of other antimicrobials was from 0.25 to 256 μg/mL. *Escherichia coli* ATCC 25922 was used as a quality control reference strain. The MDR was defined as the isolate being resistant to 3 or more antimicrobials.

ESBL-producing strains were confirmed by the double-disk diffusion test using the cephalosporins CAZ and CTX alone and in combination with clavulanic acid [16].

### 2.3. Detection of ESBL and Virulence Genes

Plasmid and genomic DNA from the 117 isolates were extracted by the plasmid Mini Kit I (OMEGA, Norcross, GA, USA) and by the direct boiling method, respectively. Plasmid DNA was used as a PCR template to detect the primary ESBL genes, including *bla*_TEM_, *bla*_SHV_, *bla*_OXA_, *bla*_CTX-M-1_, *bla*_CTX-M-9,_ and *bla*_CTX-M-2/8/25_. Genomic DNA was used as a template for amplifying 11 virulence genes, including SPIs (*invA*, *avrA*, *ssaQ*, *mgtC*, *sopB*, *sopE*), virulence plasmids (*spvR*, *spvB*, *spvC*), *stn*, and *bcfC*. The primers used to amplify ESBL and virulence genes in this study are listed in Table 1.

All of the PCR products were sequenced by TSINGKE Biological Technology (Chengdu) Co., Ltd. (Chengdu, China), and the results were analyzed using the BLAST search program 2.2 at the National Center for Biotechnology Information (NCBI) website.

### 2.4. Multilocus Sequence Typing (MLST)

MLST of all the *Salmonella* isolates was performed by the amplifying and sequencing of 7 housekeeping genes (*aroC*, *dnaN*, *hemD*, *hisD*, *purE*, *sucA*, and *thrA*) according to the MLST protocols described on the MLST website (http://mlst.warwick.ac.uk/mlst/, accessed on 1 June 2021). Alleles and STs of isolates were obtained from the *Salmonella* database according to the MLST scheme website (https://pubmlst.org/Salmonella/, accessed on 8 June 2021). The minimum spanning tree was constructed using BioNumerics 8.0 software (Applied Maths, Sint-Martens-Latem, Belgium).

### 2.5. Plasmid Conjugation

Plasmid conjugation was performed between 48 ESBL-producing isolates harboring ESBL and *spv* genes, and *E. coli* J53 AZ^r^ using the broth method [23]. The transconjugant was selected on Trypsin soybean agar medium supplemented with ampicillin (100 μg/mL) and sodium azide (200 μg/mL), and the positive transconjugant was further confirmed by the antimicrobial susceptibility test and PCR.

### 2.6. Statistical Analyses

All data entry and analyses were performed using the statistical software SPSS 25.0 and Microsoft Excel 2016. Comparisons of all variables between ESBL-producing and non-ESBL-producing isolates were carried out by the χ^2^ test. Binary logistic regression was used to analyze the relationship between the resistance phenotype and genotype, and the resistance gene and virulence gene, respectively. A value of *p* < 0.05 was considered statistically significant.

## 3. Results

### 3.1. Antimicrobial Resistance and MDR Patterns

The 117 *Salmonella* isolates showed different resistance frequencies ranging from 21.37% to 97.44%. A high rate of resistance was observed against nalidixic acid (97.44%), ampicillin (92.31%), cefazolin (73.50%), ceftriaxone (57.26%), cefotaxime (56.41%), and colistin sulfate (50.48%) (Table 2). Among all the isolates, 80 (68.38%) were multidrug-resistant strains (Figure 1).

Furthermore, 48 (41.03%) of the 117 isolates were identified as phenotypically ESBL-producing strains. Compared with the non-ESBL-producing isolates (ESBL^−^), ESBL-producing isolates (ESBL^+^) showed a higher rate of resistance to β-lactams and doxycycline, but a lower rate of resistance to gentamicin, kanamycin, and ciprofloxacin (*p* < 0.05). In addition, these ESBL^+^ isolates exhibited a higher prevalence of multidrug resistance and were mainly resistant to 6 or 7 of the antimicrobials tested (Figure 2).

### 3.2. Distribution of ESBL Genes in the ESBL^+^ and ESBL^−^ Salmonella Isolates

The 4 ESBL genes were detected in 48 ESBL^+^ and 31 ESBL^−^ isolates, with the most common being *bla*_CTX-M_, including *bla*_CTX-M-55_ (63.29%) and *bla*_CTX-M-65_ (20.25%), followed by *bla*_OXA-31_ (26.58%), and *bla*_TEM-1_ (18.99%), respectively. On the other hand, neither *bla*_CTM-2/8/25_ nor *bla*_SHV_ was detected in the test. Notably, 25.32% of the isolates carried 2 or more ESBL genes. Furthermore, the presence of *bla*_CTX-M_ in ESBL^+^ isolates (93.75%) was significantly higher than in ESBL^−^ isolates (41.94%) (*p* < 0.05), whereas the prevalence of *bla*_TEM-1_ and *bla*_OXA-31_ was higher in the ESBL^−^ isolates (*p* < 0.05) (Table 3).

### 3.3. Correlation Analysis of ESBL Genotypes and Resistance Phenotypes

The consistency between resistant phenotypes and resistance genes was calculated as previously reported [24]. The consistency between β-lactam-resistant phenotypes and ESBL genes of *Salmonella* in the study was 73.15%. Logistic regression analysis was conducted with X1 (*bla*_TEM-1_), X2 (*bla*_CTX-M-55_), X3 (*bla*_CTX-M-65_), and X4 (*bla*_OXA-31_) as independent variables and Y1 (ampicillin), Y2 (cefazolin), Y3 (ceftriaxone), and Y4 (cefotaxime) as dependent variables [25]. The resistance to cephalosporins (cefazolin, ceftriaxone, or cefotaxime) was associated with at least one ESBL gene, and *bla*_CTX-M-55_ especially showed the highest effect on the opposition to cephalosporins. However, there was no association between the resistance to ampicillin and the presence of ESBL genes and the resistance to cephalosporins and *bla*_TEM-1_. The regression equations of the compound influence on the resistance rate (P) of Y2, Y3, and Y4 were:$$P2=\frac{\exp(0.45+1.74X2)}{1+\exp(0.45+1.74X2)}$$
$$P3=\frac{\exp(-0.55+2.35X2+1.78X3-1.27X4)}{1+\exp(-0.55+2.35X2+1.78X3-1.27X4)}$$
$$P4=\frac{\exp(-1.09+3.11X2+2.26X3)}{1+\exp(-1.09+3.11X2+2.26X3)}$$

### 3.4. Prevalence of Virulence Genes

All 117 *Salmonella* isolates carried virulence genes. The prevalence of virulence genes was the highest for *invA* and *stn* (100%), followed by *sopE* (94.87%), *spvR* (87.18%), *ssaQ* (85.47%), *avrA* (77.78%), *spvB* (71.79%), *bcfC* (69.23%), *spvC* (54.70%), *sopB* (51.28%), and *mgtC* (29.06%). Among the 37 virulence gene combinations detected, the most prevalent combination was *invA-stn-avrA-ssaQ-mgtC-sopB-sopE-bcfC-spvR-spvB-spvC* (23.08%, 27/117), followed by *invA-stn-avrA-ssaQ-sopB-sopE-bcfC-spvR-spvB-spvC* (19.66%, 23/117), and *invA-stn-avrA-ssaQ-sopE-spvR-spvB* (8.55%, 10/117). In contrast, the prevalence of other combinations was lower than 6%.

Except for *invA* and *stn*, the rate of virulence genes in ESBL^+^ was higher than that of the ESBL^−^ isolates (*p* < 0.05, Figure 3). The virulence profiles of ESBL^+^ isolates were complex, and the most prevalent profile was *invA-stn-avrA-ssaQ-sopB-sopE-bcfC-spvR-spvB-spvC* (47.92%, 23/48). The ESBL and virulence genes were adopted as the independent and dependent variables, respectively. Logistic regression analysis showed that the detection rate of *avrA*, *ssaQ*, *mgtC*, *sopB*, *bcfc*, *spvR*, *spvB*, and *spvC* was associated with at least one ESBL gene, especially *bla*_CTX-M_. The emergence of ESBL genes has led to a significant increase in the detection rates of most virulence genes. In contrast, the prevalence of *sopE* was not associated with any resistance genes (Table 4).

### 3.5. MLST Analysis of Salmonella Isolates

The 117 isolates were divided into 11 STs using MLST, and the main types were ST92 (*n* = 29), ST11 (*n* = 26), and ST3717 (*n* = 22). The distribution of STs is labeled in Figure 4. The ESBL^+^ isolates were divided into 7 STs, mainly ST92 and ST3717, whereas the ESBL^−^ isolates were divided into 10 STs, mainly ST11 and ST516 (Figure 5).

The distribution of antimicrobial resistance among the 11 STs revealed differences in resistance (Figure 6). ST19, ST92, ST3717, ST2151, and ST831 showed high resistance to the third-generation cephalosporins, whereas ST831, ST516, and ST34 showed increased resistance to aminoglycoside. In addition, we found that different STs carried the same resistance genes, and *bla*_CTX-M-55_, *bla*_CTX-M-65_, *bla*_OXA-31,_ and *bla*_TEM-1_ were prevalent in different STs.

### 3.6. Transfer Ability of ESBL and spv Genes

Conjugation assay was successful in 21 ESBL^+^
*Salmonella* isolates, and these isolates were divided into 5 STs, among which ST3717 was the dominant type (Table 5). The antimicrobial susceptibility test showed that 21 transconjugants were resistant to β-lactams. In addition, it was observed that the values of MIC_50_ and MIC_90_ increased by 2–32 fold and 64–1024 fold, respectively, compared with the receptor *E. coli* J53 AZ^r^. PCR-sequencing confirmed the transfer of ESBL, and *spv* genes were found on conjugative plasmids of transconjugants. All *bla*_TEM-1_ and *bla*_CTX-M-55_ were transferred to the transconjugants; 33.33% of *bla*_CTX-M-65_ and 50.00% of *bla*_OXA-31_ were transferred to transconjugants; and the transfer rates of the *spvR*, *spvB*, and *spvC* were 71.43%, 75.00%, and 71.43%, respectively.

## 4. Discussion

In this study, we investigated the prevalence of AMR among 117 *Salmonella* isolates obtained from sick and dead chickens in the Sichuan Province, China, between 2019 and 2021. Herein, the highest resistance was observed against ampicillin (92.31%) and nalidixic acid (97.44%), which was consistent with reports from other regions of China [12,26]. Interestingly, these two agents are rarely used in the poultry industry, which suggests that the development of resistance may be related to antimicrobial activity or cross-resistance. Furthermore, compared with the third-generation cephalosporins (CRO: 57.26%; CTX: 56.41%), the resistance rate of the first-generation cephalosporin (CFZ: 73.50%) was higher, which was consistent with the results from Henan Province [27]. In addition, we observed high resistance rates to tetracycline (41.03%) and colistin sulfate (50.43%). Resistance to colistin had also been observed in central China (colistin, 51.2%) [12]. Although the prevalence of colistin-resistant clinical strains significantly decreased due to withdrawing colistin as a feed additive in 2017, the relevant analysis indicated the percentage of IncI2-type plasmids, which harbored *mcr* and other resistance genes such as *bla*_CTX-M_, significantly increased after the colistin withdrawal. The plasmids might be positively selected with the extensive use of β-lactams in food animals [28]. Notably, the rate of multidrug-resistant isolates in this study was 68.38%, which was higher than in isolates from Central China (49.5%) [12] and Shandong Province (55.8%) [26]; however, this was similar to the report in Henan Province (67.5%) [27]. The emergence of these multidrug-resistant isolates is particularly concerning because of further limits of antimicrobial selection for treating salmonellosis. It represents a significant threat to public health if these multidrug-resistant isolates are transmitted to humans through the food chain.

The production of ESBL has been identified as a mechanism of resistance to the third-generation cephalosporins. The ESBL-producing strains are highly resistant to β-lactams, and they may also be highly resistant to other antimicrobials, including aminoglycosides, tetracyclines, and quinolones [29]. Recent studies have indicated an annual increase in the prevalence of ESBL-producing *Salmonella* [12,27,29]. In this study, the prevalence of ESBL-producers (41.03%) was higher than in the isolates previously reported from different sources (usually less than 30%) [30,31]. The analysis of resistance in this study showed that ESBL-producers had higher β-lactams and doxycycline resistance, but lower resistance to aminoglycosides and ciprofloxacin than non-ESBL-producers. The latter result was inconsistent with other studies [12,27,29]. Therefore, the selection of antimicrobial agents should be prudent and based on the results of antimicrobial susceptibility tests when treating salmonellosis caused by ESBL-producing strains.

The 4 ESBL genes were detected in 79 *Salmonella* isolates (including 48 ESBL^+^ and 31 ESBL^−^ isolates). The *bla*_CTX-M-55_ (63.29%) was the most prevalent gene, which was consistent with the results previously detected in *Salmonella* from patients and food-animals in China [32], in which *bla*_CTX-M-65_ (20.25%) was first detected in *Salmonella* from chickens in Sichuan. With third-generation cephalosporins widely used in veterinary medicine, *bla*_CTX-M_ variants are evolving rapidly. Enzyme variants such as CTX-M-55 and CTX-M-65 are increasingly being detected in isolates from food animals and demonstrate high-levels of resistance to extended-spectrum cephalosporins and often demonstrate cross-resistance to other antimicrobials. *Salmonella* harboring these *bla*_CTX-M_ variants can cause clinical and food safety problems, highlighting the need to investigate the prevalence of these isolates in the livestock industry [32]. Furthermore, *bla*_OXA-31_ (26.58%) was this study’s second most common ESBL gene. Previous studies have shown that early OXA β-lactamases confer resistance to narrow-spectrum penicillins, whereas OXA-type ESBL such as OXA-31 confers resistance to extended-spectrum cephalosporins. These new *bla*_OXA_ variants could be widely spread via plasmids or integrons [33], which highlights the need for increased monitoring of *bla*_OXA_-positive microbes in food animals. In this study, only *bla*_TEM-1_ (18.99%) was detected. Unlike TEM-type ESBL, TEM-1 only confers resistance to penicillins and early cephalosporins. However, studies have shown that *bla*_TEM-1_ can easily mutate and has stable transmission ability through plasmids. Over 170 variants of TEM-1 have been isolated worldwide, and TEM-1 variants have a wide range of hydrolytic substrates such as modern β-lactams [34]. Notably, in the present study, multiple ESBL genes were simultaneously detected in the same isolate, leading to high levels of resistance and multidrug-resistant isolates. Statistical analyses showed that the consistency between ESBL genes and β-lactam resistance was 73.15%, where *bla*_CTX-M-55_ played a key role in resistance.

In general, the pathogenicity of *Salmonella* depends on the interaction of a large number of virulence factors, including the type III secretion systems encoded on SPIs, virulence plasmid(*spv*) factors, enterotoxin, and structural virulence factors (fimbriae and flagella) [11]. The ability of nontyphoidal *Salmonella* to cause invasive disease is attributed to the SPIs genes. The *invA*, *avrA*, *ssaQ*, *mgtC*, *siiD*, and *sopB* genes belong to SPI 1–5 in *Salmonella* [35]. The present study showed that except for *mgtC* (29.06%), the detection rate of SPIs tested was high (ranging from 51.28%to 100%), indicating these virulence genes were widespread and highly conserved. The *spvABCD* system and its regulator *spvR* are the key factors in enhancing the transmission and reproductive capacity of *Salmonella* in the reticuloendothelial systems, and are associated with extra-intestinal infections [11]. The existence of *spv* and other virulence genes (*stn*, *bcfC*) tested was also high (ranging 54.70–100%). Based on the virulence profiles, there were 37 combinations of the virulence genes. Importantly, ESBL^+^ isolates had a higher prevalence of virulence genes and more complex profiles than ESBL^−^ isolates. We explored the relationship between the resistance and virulence genes through logistic regression analysis. We found that the prevalence of *avrA*, *ssaQ*, *mgtC*, *sopB*, *bcfC*, *spvR*, *spvB*, and *spvC* was positively influenced by at least one ESBL gene. In addition, the presence of the *bla*_CTX-M_ gene significantly increased the presence of *spv* genes, whereas no relation was observed between *sopE* and resistance genes.

MLST is used to distinguish *Salmonella* subtypes, and it can break through the limitations of traditional serotyping. In this study, MLST was used to identify each strain using seven housekeeping genes. The results showed that the 117 *Salmonella* isolates were divided into 11 STs, among which ST92 (24.79%) was the most prevalent. This result was consistent with the report on *Salmonella* STs of chicken from nine provinces in China [36]. Furthermore, the ESBL^+^ isolates had seven STs (mainly ST 92 and ST 3717), whereas ESBL^–^ had 10 STs (mainly ST11 and ST 516). In analyzing the relationship between the STs and resistance phenotype, five STs of isolates (including ST19, ST92, ST37117, ST2151, and ST831) showed high resistance to the third generation cephalosporins, whereas three STs (ST831, ST516, and ST34) showed high aminoglycoside resistance.

Plasmids are an important vector for capturing, accumulating, and transmitting antimicrobial resistance genes. The mobilizable DNA fragments in plasmids, such as resistance genes and virulence genes, can be transferred by conjugation from a donor to a recipient cell [37]. Twenty-one transconjugants obtained in our study showed different resistance to four β-lactam antimicrobials. Furthermore, 71.43% of transconjugants presented both ESBL and *spv* genes, suggesting that the co-transfer of *spv* genes and ESBL genes dominated by *bla*_CTX-M_ occurred. Previous studies [38] have reported that the *spv* genes were located on a self-transmissible virulence plasmid that carried resistance genes, highlighting the high risk of spreading virulent strains that harbored ESBL genes. Therefore, further resistance surveillance must be focused on the interaction between resistance and pathogenicity.

## 5. Conclusions

This study reveals high occurrence of multidrug-resistant and ESBL-producing *Salmonella* in chickens in the Sichuan Province of China, and highlights that ESBL-producing *Salmonella* isolates carried various virulence genes. In addition, it was found that ESBL and virulence genes could be co-transmitted to other strains through plasmid transfer. This not only reveals transmission mechanisms of resistance genes and virulence genes, but also highlights the potential danger of *Salmonella* epidemics that carry genes for drug resistance and virulence. These results point to a significant health risk in this region, which may result in severe clinical infection and loss of productivity, and even threaten human health. Together, these findings emphasize the importance of continuously monitoring antibiotics in the animal industries and adopting public health measures to contain the spread of *Salmonella* isolates resistant to the critical antibiotics employed for the treatment of severe salmonellosis.

## Figures and Tables

**Figure 1 animals-13-02770-f001:**
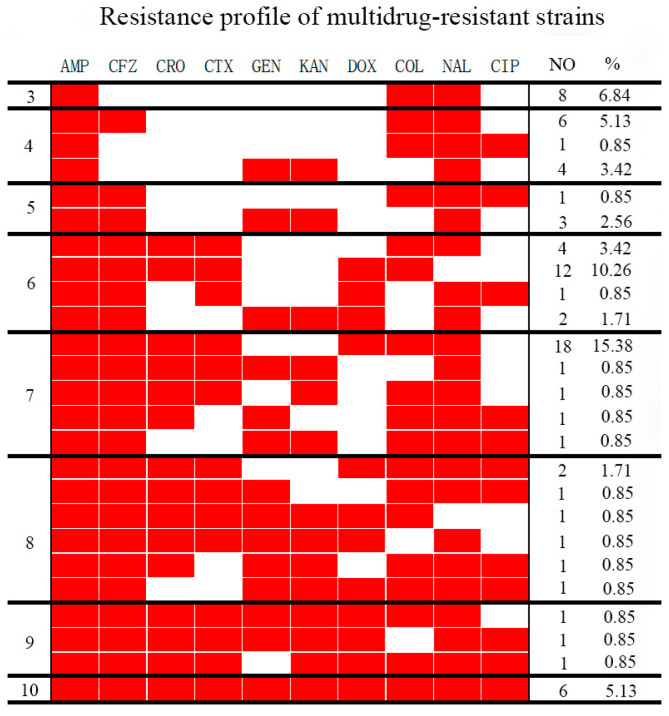
The rate of multidrug-resistant *Salmonella* isolates. AMP, ampicillin; CFZ, cefazolin; CRO, ceftriaxone; CTX, cefotaxime; GEN, gentamicin; KAN, kanamycin; DOX, doxycycline; NAL, nalidixic acid; CIP, ciprofloxacin; COL, colistin sulfate; 3—resistance to 3 antimicrobials; 4—resistance to 4 antimicrobials; 5—resistance to 5 antimicrobials; 6—resistance to 6 antimicrobials; 7—resistance to 7 antimicrobials; 8—resistance to 8 antimicrobials; 9—resistance to 9 antimicrobials; 10—resistance to 10 antimicrobials; red, resistance; white, intermediate or susceptible.

**Figure 2 animals-13-02770-f002:**
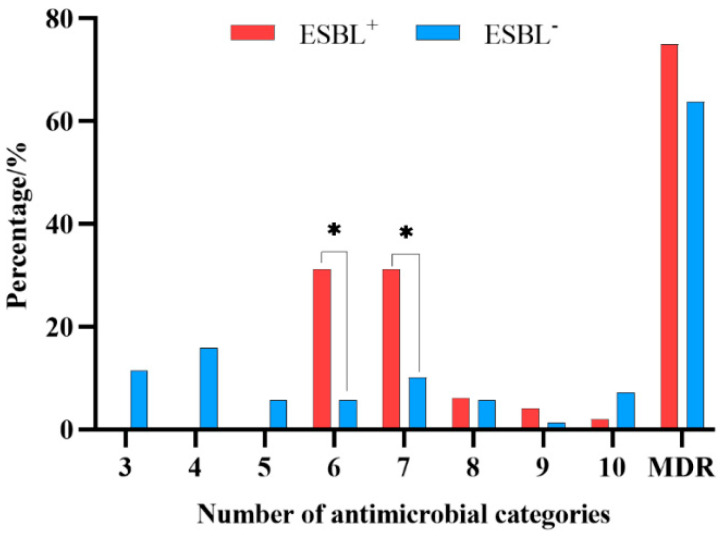
Comparison of multidrug-resistance rate between ESBL^+^ and ESBL^−^ isolates. 3—resistance to 3 antimicrobials; 4—resistance to 4 antimicrobials; 5—resistance to 5 antimicrobials; 6—resistance to 6 antimicrobials; 7—resistance to 7 antimicrobials; 8—resistance to 8 antimicrobials; 9—resistance to 9 antimicrobials; 10—resistance to 10 antimicrobials; ESBL^+^ vs. ESBL^−^ multidrug-resistance rate, * *p* < 0.05.

**Figure 3 animals-13-02770-f003:**
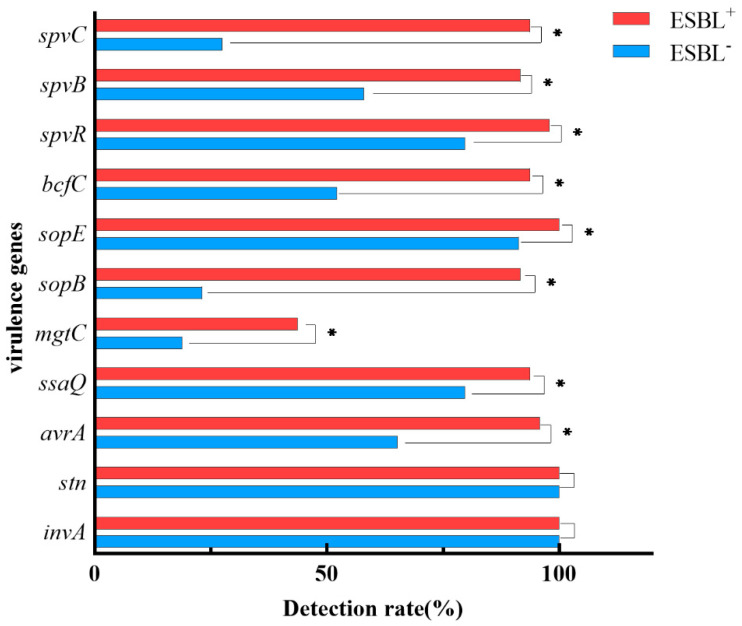
The detection of virulence genes in ESBL^−^ and ESBL^+^ isolates. The detection rates of virulence genes in ESBL^+^ vs. ESBL^−^, * *p* < 0.05.

**Figure 4 animals-13-02770-f004:**
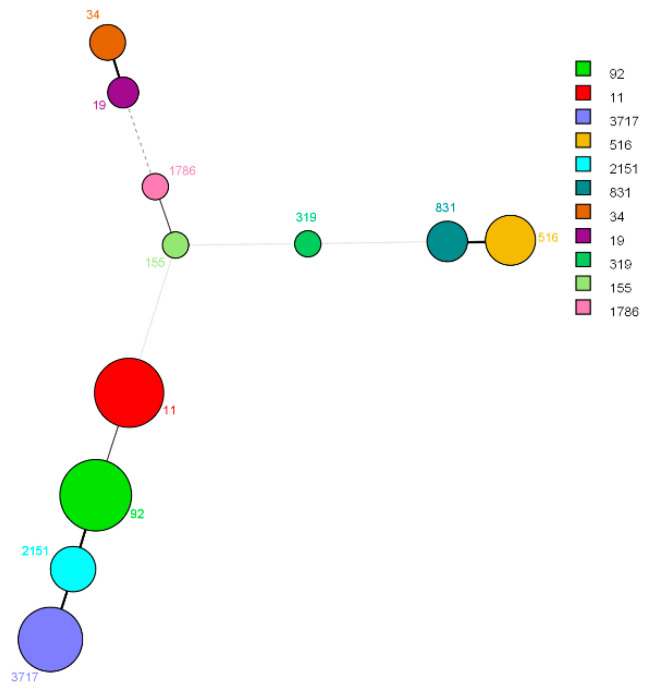
Minimum spanning tree analysis of 117 *Salmonella* isolates based on allelic profiles of 7 housekeeping genes. Each circle corresponds to a ST and the number of each ST is given beside the circle. The circle size depends on the number of isolates found within that profile, and the width of the line connecting 2 strains indicates the genetic variation in these isolates.

**Figure 5 animals-13-02770-f005:**
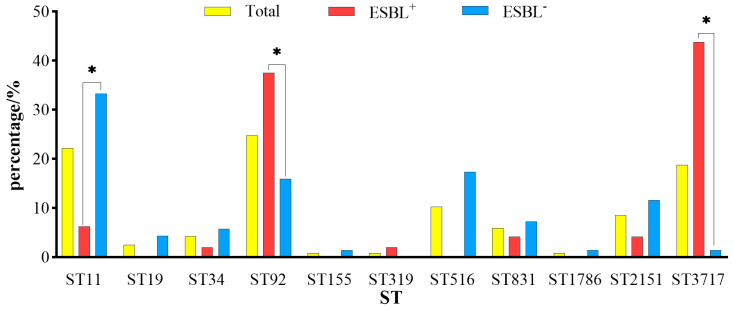
The occurrence of ST among the ESBL^−^ and ESBL^+^ isolates. The detection rates of ST in ESBL^+^ vs. ESBL^−^, * *p* < 0.05.

**Figure 6 animals-13-02770-f006:**
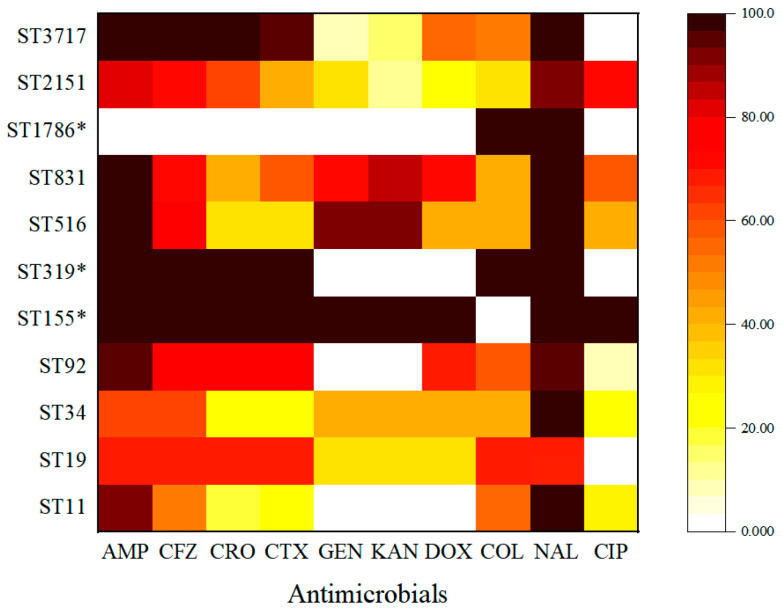
The distribution of AMR in different STs of *Salmonella* isolates. The distribution of the average antimicrobial resistance (in percent) of various STs towards 10 antimicrobials. The color of individual cells varies with the percentage of antimicrobial resistance shown in the cells. * ST containing only one strain.

**Table 1 animals-13-02770-t001:** Primers for ESBL and virulence genes of *Salmonella*.

Target Genes	Sequence (5′-3′)	Product Size (bp)	Annealing Temperature (°C)	Reference
*bla* _TEM_	F:ATAAAATTCTTGAAGACGAAA R:GACAGTTACCAATGCTTAATC	1080	55	[17]
*bla* _SHV_	F:TTATCTCCCTGTTAGCCACC R:GATTTGCTGATTTCGCTCGG	797	57	[17]
*bla*_CTX-M-1_ group	F:GGTTAAAAAATCACTGCGTC R:TTACAAACCGTCGGTGACGA	874	55	[17]
*bla*_CTX-M-9_ group	F:AGAGTGCAACGGATGATG R:CCAGTTACAGCCCTTCGG	868	57	[17]
*bla*_CTX-M-2/8/25_ group	F:ACCGAGCCSACGCTCAA R:CCGCTGCCGGTTTTATC	221	60	[17]
*bla_OXA_*	F:AATGGCACCAGATTCAACTT R:CTTGGCTTTTATGCTTGATG	595	60	[18]
*invA*	F:GTGAAATTATCGCCACGTTCGGGCAA R:TCATCGCACC GTCAAAGGAACC	284	55	[15]
*avrA*	F:GTTATGGACGGAACGACATCGG R:ATTCTGCTTCCCGCCGCC	389	60	[19]
*ssaQ*	F:GAATAGCGAATGAAGAGCGTCC R:CATCGTGTTATCCTCTGTCAGC	677	60	[15]
*mgtC*	F:TGACTATCAATGCTCCAGTGAAT R:ATTTACTGGCCGCTATGCTGTTG	655	60	[15]
*sopB*	F:GATGTGATTAATGAAGAAATGCC R:GCAAACCATAAAAACTACACTCA	1170	60	[15]
*sopE*	F:CGAGTAAAGACCCCGCATAC R:GAGTCGGCATAGCACACTCA	362	58	[20]
*spvR*	F:AGGAAATCGGACCTACGG R:TAACATCGCCAGCCCTTG	473	57	[15]
*spvB*	F:CCTGATGTTCCACCACTTTC R:ATGCCTTATCTGGCGATGT	590	57	[15]
*spvC*	F:ACTCCTTGCACAACCAAATGCGGA R:TGTCTCTGCATTTCGCCACCATCA	571	56	[15]
*stn*	F:CTTTGGTCGTAAAATAAGGCG R:TGCCCAAAGCAGAGAGATTC	260	55	[21]
*bcfC*	F:ACCAGAGACATTGCCTTCC R:TTCTGATCGCCGCTATTCG	467	57	[22]

**Table 2 animals-13-02770-t002:** Frequencies of antimicrobial resistance (AMR) among ESBL^−^/ESBL^+^ and all isolates.

	Resistance Rate (%)		
Antimicrobials	ESBL^−^ (*n* = 69)	ESBL^+^ (*n* = 48)	All Isolates (*n* = 117)
Ampicillin	86.96	100.00 *	92.31
Cefazolin	55.07	100.00 *	73.50
Ceftriaxone	27.54	100.00 *	57.26
Cefotaxime	26.09	100.00 *	56.41
Gentamicin	31.88	6.25 *	21.37
Kanamycin	28.99	10.42 *	21.37
Doxycycline	23.19	66.67 *	41.03
Colistin sulfate	52.17	47.92	50.43
Nalidixic acid	95.65	100.00	97.44
Ciprofloxacin	31.88	10.42 *	23.08

Note: *t*-test, comparing differences between ESBL^−^ and ESBL^+^ resistance frequencies. * *p* < 0.05.

**Table 3 animals-13-02770-t003:** The distributions of ESBL genes in the ESBL^−^ and ESBL^+^ isolates.

Isolates	*bla*_TEM-1_ (%)	*bla*_CTX-M_ (%)	*bla*_OXA-31_ (%)
ESBL^−^	38.71 (12/31)	41.94 (13/31)	48.39 (15/31)
ESBL^+^	6.25 (3/48) *	93.75 (45/48) *	12.50 (6/48) *
Total	18.99 (15/79)	73.42 (58/79)	26.58 (21/79)

Note: *t*-test, comparing differences between the distributions of ESBL genes in the ESBL^−^ and ESBL^+^ isolates. * *p* < 0.05.

**Table 4 animals-13-02770-t004:** Logistics regression results of ESBL genes and virulence genes.

	Single Factor Effect	Multiple Factors Effect	No Significant Effect
*avrA*		*bla*_CTX-M-55_, *bla*_OXA-31_, *bla*_TEM-1_	*bla* _CTX-M-65_
*ssaQ*	*bla* _CTX-M-55_		*bla*_CTX-M-65_, *bla*_OXA-31_, *bla*_TEM-1_
*mgtC*	*bla* _CTX-M-65_		*bla*_CTX-M-55_, *bla*_OXA-31_, *bla*_TEM-1_
*sopB*		*bla*_CTX-M-55_, *bla*_CTX-M-65_, *bla*_OXA-31_, *bla*_TEM-1_	
*sopE*			*bla*_CTX-M-55_, *bla*_OXA-31_, *bla*_CTX-M-65_, *bla*_TEM-1_
*bcfC*		*bla*_CTX-M-55_, *bla*_CTX-M-65_	*bla*_OXA-31_, *bla*_OXA-31_
*spvR*	*bla* _CTX-M-55_		*bla*_OXA-31_, *bla*_CTX-M-65_, *bla*_TEM-1_
*spvB*	*bla* _CTX-M-55_		*bla*_OXA-31_, *bla*_CTX-M-65_, *bla*_TEM-1_
*spvC*		*bla*_CTX-M-55_, *bla*_CTX-M-65_	*bla*_OXA-31_, *bla*_TEM-1_

**Table 5 animals-13-02770-t005:** The results of conjugation transfer and STs of 21 donors and their transconjugants.

Isolates	STs	Donor Bacterium		Transconjugants	
		Resistance and Virulence Genes	Resistant Phenotype	Resistance and Virulence Genes	Resistant Phenotype
SLZC19-128	ST92	*bla*_CTX-M-55_, *spvR*, *spvB*, *spvC*	AMP, CFZ, CRO, CTX	*bla* _CTX-M-55_	AMP
SLZC19-605	ST92	*bla*_CTX-M-55_, *bla*_CTX-M-65_, *spvR*, *spvB*, *spvC*	AMP, CFZ, CRO, CTX	*bla* _CTX-M-55_	AMP, CFZ, CRO, CTX
SLZC19-628	ST92	*bla*_CTX-M-55_, *bla*_CTX-M-65_, *bla*_OXA-31_, *spvR*, *spvB*, *spvC*	AMP, CFZ, CRO, CTX	*bla* _CTX-M-55_	AMP
SLZC20-52	ST11	*bla*_CTX-M-65_, *bla*_OXA-31_, *spvR*, *spvB*, *spvC*	AMP, CFZ, CRO, CTX	*bla*_CTX-M-65_, *bla*_OXA-31_	AMP, CFZ, CRO, CTX
SLZC20-53	ST11	*bla*_CTX-M-65_, *bla*_OXA-31_, *spvR*, *spvB*, *spvC*	AMP, CFZ, CRO, CTX	*bla* _CTX-M-65_	AMP, CFZ, CRO, CTX
SDZC21-1	ST3717	*bla*_CTX-M-55_, *spvR*, *spvB*, *spvC*	AMP, CFZ, CRO, CTX	*bla*_CTX-M-55_, *spvR*, *spvB*, *spvC*	AMP, CFZ, CRO, CTX
SDZC21-3	ST3717	*bla*_CTX-M-55_, *spvR*, *spvB*, *spvC*	AMP, CFZ, CRO, CTX	*bla*_CTX-M-55_, *spvR*, *spvB*, *spvC*	AMP, CFZ, CRO, CTX
SDZC21-4	ST3717	*bla*_CTX-M-55_, *bla*_CTX-M-65_, *spvR*, *spvB*, *spvC*	AMP, CFZ, CRO, CTX	*bla*_CTX-M-55_, *spvR*, *spvB*, *spvC*	AMP, CFZ, CRO, CTX
SDZC21-5	ST3717	*bla*_CTX-M-55_, *spvR*, *spvB*, *spvC*	AMP, CFZ, CRO, CTX	*bla*_CTX-M-55_, *spvR*, *spvB*, *spvC*	AMP, CFZ, CRO, CTX
SDZC21-8	ST3717	*bla*_TEM-1_, *bla*_CTX-M-55_, *spvR*, *spvB*, *spvC*	AMP, CFZ, CRO, CTX	*bla*_TEM-1_, *bla*_CTX-M-55_, *spvR*, *spvB*, *spvC*	AMP, CFZ, CRO, CTX
SDZC21-10	ST3717	*bla*_CTX-M-55_, *spvR*, *spvB*, *spvC*	AMP, CFZ, CRO, CTX	*bla*_CTX-M-55_, *spvR*, *spvB*, *spvC*	AMP, CFZ, CRO, CTX
SDZC21-11	ST3717	*bla*_CTX-M-55_, *spvR*, *spvB*, *spvC*	AMP, CFZ, CRO, CTX	*bla*_CTX-M-55_, *spvR*, *spvB*, *spvC*	AMP, CFZ, CRO, CTX
SDZC21-12	ST3717	*bla*_CTX-M-55_, *spvR*, *spvB*, *spvC*	AMP, CFZ, CRO, CTX	*bla*_CTX-M-55_, *spvR*, *spvB*, *spvC*	AMP, CFZ, CRO, CTX
SDZC21-13	ST3717	*bla*_CTX-M-55_, *spvR*, *spvB*, *spvC*	AMP, CFZ, CRO, CTX	*bla*_CTX-M-55_, *spvR*, *spvB*, *spvC*	AMP, CFZ, CRO, CTX
SDZC21-14	ST3717	*bla*_CTX-M-55_, *spvR*, *spvB*, *spvC*	AMP, CFZ, CRO, CTX	*bla*_CTX-M-55_, *spvR*, *spvB*, *spvC*	AMP, CFZ, CRO, CTX
SDZC21-15	ST3717	*bla*_CTX-M-55_, *spvR*, *spvB*, *spvC*	AMP, CFZ, CRO, CTX	*bla*_CTX-M-55_, *spvR*, *spvB*, *spvC*	AMP, CFZ, CRO, CTX
SDZC21-16	ST3717	*bla*_CTX-M-55_, *bla*_CTX-M-65_, *spvR*, *spvB*, *spvC*	AMP, CFZ, CRO, CTX	*bla*_CTX-M-55_, *spvR*, *spvB*, *spvC*	AMP, CFZ, CRO, CTX
SDZC21-18	ST3717	*bla*_CTX-M-55_, *spvR*, *spvB*, *spvC*	AMP, CFZ, CRO, CTX	*bla*_CTX-M-55_, *spvR*, *spvB*, *spvC*	AMP, CFZ, CRO, CTX
SDZC21-19	ST3717	*bla*_CTX-M-55_, *spvR*, *spvB*, *spvC*	AMP, CFZ, CRO, CTX	*bla*_CTX-M-55_, *spvR*, *spvB*, *spvC*	AMP, CFZ, CRO, CTX
SDZC21-26	ST155	*bla*_CTX-M-55_, *spvR*, *spvB*, *spvC*	AMP, CFZ, CRO, CTX	*bla*_CTX-M-55_, *spvR*, *spvB*, *spvC*	AMP, CFZ, CRO, CTX
SYBC20-27	ST2151	*bla*_TEM-1_, *bla*_OXA-31_, *spvR*, *spvC*	AMP, CFZ, CRO, CTX	*bla*_TEM-1_, *bla*_OXA-31_	AMP, CFZ

## Data Availability

The data presented in this study are available upon request from the corresponding author.

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
