# Peer review of "Association between Phenotypes of Antimicrobial Resistance, ESBL Resistance Genes, and Virulence Genes of Salmonella Isolated from Chickens in Sichuan, China"

_animals, 2023, doi:10.3390/ani13172770_

Round 1
Reviewer 1 Report
This manuscript investigated the association between antibiotic resistance, ESBL gene, and virulence gene of Salmonella isolates. The research topic and content of the manuscript are very interesting, but there are several errors in the manuscript that need to be modified. The specific details are as follows.
Specific comments
1. Line 15. “Salmonella” must be written by italic type. Please check the entire manuscript and modifed it.
2. Line 27. Please define the abbreviation of AMR.
3. Line 94. Please define the abbreviation of MDR.
4. Line 99-101. “Sahan et al. reported that out of …” → “Sahan et al. [4] reported that out of …”
5. Line 104. “ESBLs …” → “The ESBLs …” / Do not start sentences with abbreviations. Please check if there is a sentence starting with an abbreviation in the entire manuscript.
6. Line 105. The font size for “cephalosporins” is different from the others. Please unify the font size in the manuscript according to the journal format.
7. Line 117-119. It is not clear what the meaning of infection by MDR pathogens was written in that paragraph. A supplementary explanation is required.
8. Line 133. Is “SPIS genes” the “SPIs” defined in Line 123? If yes, please correct it.
9. Line 137. Please more explain in detail the specific hypotheses established at the beginning of this study and the purpose of this study.
10. Line 152. Please define the abbreviation of PCR.
11. Line 192-193. “100ug/mL” → “100 ug/mL” / Except for “%” and “℃”, all units must be written after a space.
12. Line 250. “reported[24]” → “reported [24]”
13. Line 253. “X3(blactx…)” → “X3 (blactx…”) / Please check for spacing errors in the manuscript and correct them all.
14. Change the page numbers and line numbers again. Since the line number is reset after page 14 now, it needs to be set again.
15. Reference 14. Please rewrite the reference properly.
16. Lastly, please check that there are no typos or grammatical errors in the manuscript.
Please check that there are no typos or grammatical errors in the manuscript.
Reviewer 2 Report
This is an interesting article regarding details of Salmonella spp., isolated from chickens in China. It is well executed and scientifically rigorous. There are however a number of issues regarding the grammatical writing which I have tried to detail below as English may not be the authors first language. I hope that its useful for the authors.
Line 15, 18, 20, 25, 29, 40, 77, 155, - Salmonella needs to be in italics and capitalised throughout
Line 17- In recent years …. (reword)
Line 21- space needed between chromosomal and the brackets
Line 22- plasmids (reword)
Line 30- resistance to antimicrobials (reword)
Line 31- ESBL producers …. (reword)
Line 39- observed in the plasmid … (reword)
Line 99- worse scenario doesn’t quite sound right- please reword
Line 100- from broiler chickens and cattle. ..(reword)
Line 101- was an ESBL producer .. (reword)
Line 109- shown that plasmids are important …. (reword)
Line 118- with high morbidity …(reword)
Line 132- up to 100% is meaningless- please add in the values
Line 140- pathogenicity of the strain… (reword)
Section 2.2.- please add in the concentration of the antibiotics
Section 2.2.- Kirby Bauer disc diffusion for colistin is poor due to its large size. There is now a new technique for this, was this followed? If so please detail it
Line 169- as a template for … (reword)
Line 174- were these sequences put onto Genbank? If so, what accession numbers are they?
Line 218- antimicrobial classes- 3 or more antimicrobial classes that the bacteria are resistant to means that they are MDR. This is also true for the legend of figure 1 and 2
Figure 4 legend- Salmonella needs capitalising.
What is the key for figure 4?
Line 68- page 2 discussion – spread via plasmids or integrons …. (reword)
Line 78- serious MDR- perhaps delete serious?
Line 105- correlation with the blaCTX-M gene … (reword)
Line 109- In this study, MLST was used …. (reword)
Line 122- no correlation between ST types … (reword)
Line 131- reported that the spv gene …. (reword)
These are detailed above
Round 2
Reviewer 1 Report
Thanks for the correction
Thanks for the correction
